# Assessing the Cooling and Air Pollution Tolerance among Urban Tree Species in a Tropical Climate

**DOI:** 10.3390/plants11223074

**Published:** 2022-11-13

**Authors:** Arerut Yarnvudhi, Nisa Leksungnoen, Tushar Andriyas, Pantana Tor-Ngern, Aerwadee Premashthira, Chongrak Wachrinrat, Dokrak Marod, Sutheera Hermhuk, Sura Pattanakiat, Tohru Nakashizuka, Roger Kjelgren

**Affiliations:** 1Department of Forest Biology, Faculty of Forestry, Kasetsart University, Bangkok 10900, Thailand; 2Center for Advance Studies in Tropical Natural Resources, National Research University-Kasetsart University, Kasetsart University, Bangkok 10900, Thailand; 3Kasetsart University Research and Development Institute (KURDI), Kasetsart University, Bangkok 10900, Thailand; 4Department of Environmental Science, Faculty of Science, Chulalongkorn University, Bangkok 10330, Thailand; 5Water Science and Technology for Sustainable Environment Research Group, Chulalongkorn University, Bangkok 10330, Thailand; 6Department of Agricultural and Resource Economics, Faculty of Economics, Kasetsart University, Bangkok 10900, Thailand; 7Department of Silviculture, Faculty of Forestry, Kasetsart University, Bangkok 10900, Thailand; 8Cooperation Centre of Thai Forest Ecological Research Network, Kasetsart University, Bangkok 10900, Thailand; 9Faculty of Agricultural Production, Maejo University, Chiang Mai 50290, Thailand; 10Faculty of Environment and Resource Studies, Mahidol University, Nakhon Pathom 73170, Thailand; 11Forest and Forest Products Research Institute, Tsukuba 300-1244, Japan; 1212HE UF/IFAS Dept. Environmental Horticulture, University of Florida, Apopka, FL 32703, USA

**Keywords:** ecosystem services, air pollution tolerance, anticipated performance index, shading

## Abstract

We present the results of classifying plants at species level that can tolerate air pollution, provide cooling, and simultaneously survive and thrive in urban environments. For this purpose, we estimated the air pollution tolerance index (APTI) and anticipated performance index (API) of several species growing in a park located in central Bangkok, Thailand. The cooling effect was quantified by calculating the reduction in soil and air temperatures. *Melaleuca quinquenervia* (Cav.) S.T. Blake, *Albizia saman* (Jacq.) Merr., *Chukrasia tabularis* A. Juss. had the highest API score and were able to substantially reduce the temperature and were in a group of highly recommended species which also included other species like *A*. *saman*, *C*. *tabularis*, *Tabebuia rosea* (Bertol.) Bertero ex A. DC., *Dalbergia cochinchinensis* Pierre etc. Species from both evergreen and deciduous habitat were able to provide ambient cooling but were vulnerable to air pollution and included *Elaeocarpus grandifloras* Sm. and *Bauhinia purpurea* L. However, there were other species which had a high air pollution tolerance but failed to provide adequate cooling, such as *Hopea odorata* Roxb. and *Millingtonia hortensis* L.f. The results would be of interest to urban greenspace landscapers in such climates while selecting suitable species that can provide multiple ecosystem services ranging from air pollution tolerance to temperature reduction without reducing plant vitality.

## 1. Introduction

Rapid urbanization and anthropogenic activities can create environmental issues in cities [1]. Such issues include urban heat islands (UHIs), when solar radiation is trapped through heating of dry surfaces like pavements and buildings, with the heat being either reradiated by long wave radiation or conducted through air movement. Additionally, industrial and greenhouse emissions from air conditioning and traffic [2] also contribute to increasing the ambient temperatures. The mean annual air temperatures in Bangkok city, Thailand has been reported to be up to 0.8 °C higher than places outside the city [3], with high UHI effects being observed in places of public transport (airports), dense residential areas, and industrial areas, while public parks and green spaces have been reported to be relatively cooler [4].

Anthropogenic activity not only causes UHIs but can also result in air pollution, especially from construction, industrial production, agricultural stubble burning, traffic, and cars, increasing the levels of particulate matter (PM) [5], especially during the cool dry season in monsoonal regions caused by the air inversion effect [2]. Both UHI and air pollution can affect human health as well as plant growth and vitality.

Mitigation is needed to reduce the effects of UHI and air pollution in urban areas. Specific plant species and their ecosystem services can be used to regulate the microclimate by removing air pollution [6] and reducing ambient temperatures through transpirational cooling and shading [7]. Air pollution tolerant index (APTI) and anticipated performance index (API) can be used to quantify and classify the level of air pollution tolerance of various plant species [8,9]. Plants can help to reduce the ambient temperatures through transpiration, acting as a natural air conditioner, providing valuable ecosystem services for humans [10]. The U.S. Environmental Protection Agency [11] reported that shading from plants reduced the peak air temperature in summer by 1–5 °C while Shiflett, et al. [12] reported reduction within a range of 3–6 °C in the Greater Los Angeles metropolitan area. Cooling through plant transpiration is more pronounced for deciduous species due to relatively large leaves [13] in addition to dense and large canopies, with such characteristics being positively related with the leaf area index (LAI) [14]. Canopy characteristics can vary based on species, age, and location, affecting the air and soil temperature levels under the canopy [15].

For plants to be able to provide ecosystem services related to air pollution tolerance and cooling, they must be healthy and able to survive in such urban environments through structural, physiological, metabolic, and biochemical defense mechanisms [16]. Species with hairs and a thick waxy coating are the effective characteristics to accumulate PM. Trees in the center of the park would retain PM in longer time when compare to those at the edge or close to the road because of dense tall canopy acting as the insulting layer for air pollution [17]. Leaves are generally used to evaluate the sensitivity of plants to air pollutants as they are in direct contact with the ambient air. As such, plant leaves have been recommended to determine their ability to absorb and/or adsorb the pollutants [18,19]. The uptake of gaseous air pollutants is primarily done via leaf stomata, though some gases are removed by the plant surface [20]. Some PM can be retained on the leaf surface and often re-suspended in the atmosphere by wind, washed away by rain, or dropped to the ground during leaf and twig fall [21]. Plants also experience stress caused by high pollution concentration, reducing the plant’s ability to photosynthesize efficiently [22]. As a result, the overall plant growth and development can be severely affected by leaf damage [23].

Under stresses induced by air pollution, a highly acidic (low pH) environment can build up in the cell sap due to acidic gases (SO_2_, NO_2_, or CO_2_) in the ambient air forming acid radicals in the leaf tissue [24,25]. Such acidic gases can damage the cell membrane after entering through the stomata [26]. Therefore, species that can maintain alkaline levels would be more tolerant to air pollution. Relative water content (R) is associated with cell turgor pressure and protoplasmic permeability [27] with high level diluting acidity inside the leaf cell sap, resulting in resistance to air pollution [27,28,29]. Kumar and Nandini [28] indicated that stress caused by air pollution can decrease the chlorophyll content in species. As chlorophyll content is directly related to the photosynthesis process and in turn to the growth and development of plants, a high chlorophyll content would be directly related to high growth and air pollution tolerance.

Selecting viable plant species that can survive in urban environments and provide ecosystem services is very important as only healthy plants can provide optimum services without compromising plant’s health. To our knowledge, only a limited number of studies have been done in the Asian tropical monsoon climates focusing on plant species tolerant to air pollution as well as providing ambient cooling. Only one study related to APTI in the northern part of Thailand has been reported till date [30], while the cooling effects at the individual species level have been evaluated only for a limited number of species (with *A. saman*) in an urban park setting in Bangkok, Thailand [31].

The purpose of this study was to classify the trees at species level based on their relative tolerance to air pollution and cooling, which can simultaneously survive and thrive in urban environments based on their APTI and API values. Species were sampled in an urban green space (known as CU100 Park), established in central Bangkok, Thailand. This Park was designed to prevent inland flooding and to make the surrounding area more resilient to climate change [6].

## 2. Results

The selected species were classified as either evergreen (10 species) or deciduous (11 species) (Table 1). *D. cochinchinensis* was the most frequently found species (136 individuals), while *E. grandiflorus* (5 individuals) was the least frequently recorded. The DBH ranged between 7.86–19.70 cm with an average of 13.62 ± 2.27 cm, while the height was measured between 3.28–9.70 m with an average of 7.64 ± 2.22 m. *A. saman* had the largest DBH while *E. grandiflorus* was the smallest and the shortest in this study. *A. saman* also had the largest crown size while *Dipterocarpus alatus Roxb. ex G. Don* had the smallest crown cover. *Afzelia xylocarpa (Kurz) Craib* had the densest canopy while *Terminalia bellirica (Gaertn.) Roxb.* had the most open crown canopy (Table 2).

Low to moderate pollution levels (Table 3) were observed for the 21 species as indicated by an APTI ranging from intermediate to tolerable with values between 10.35 to 13.06 (Table 4). *A. saman* had the highest APTI (Table 4) given the highest measured ascorbic acid, pH, and total chlorophyll content (data not shown). Moreover, its hairy leaves can trap particulate matter (PM_10_ or PM_2.5_), whose levels are the most problematic in the urban areas of Thailand, especially during the cool dry months of the year (December to February), when the area experiences an air inversion. *M. hortensis* and *M. quinquenervia* had the next highest levels of APTI. APTI can only explain how tolerant each individual plant is to pollutants, but the API is used as a criterion to recommend the planting suitability of a certain species in urban environments. API was calculated based on APTI, biochemical characteristic, plant size, canopy structure, type of plant, leaf structure, and economic value (Table 2). The API ranged from 3–7, with most species (14 species) in this Park falling in the moderate to good category, with API scores 4 (Table 4). All the recommended species were evergreen, with a very good to the best API score between 5–7. The most recommended species with the highest API score was *M. quinquenervia*, given that it had a high APTI and of excellent economic value (Table 4). Additionally, being an evergreen species, it can provide shade all year round, with a well-rounded dense canopy structure. Its hairy and tough small leaf structure results in a high pollution absorption capacity. Based on the existing conditions and the air pollution levels, the recommended species to be planted (with a score of 5) in this park were *A. saman*, *C. tabularis*, *Artocarpus lacucha* Roxb. ex Buch.-Ham., and *Saraca indica* L. In contrast, *Homalium tomentosum* (Vent.) Benth. was the least recommended (with score of 3), given its leaf structure being big, of smooth texture, and fragile resulting in a relatively lower capacity to repel pollutants.

Reduction in the annually averaged soil temperature (ΔTs), calculated for the 21 selected species, ranged from 3.56 °C to 5.90 °C (Table 5). The exact positon of trees in the park was shown in the Figure 1. Most of the trees were distant from the road in approximately the same (as in Figure 1, the main road are around the park). The density of the trees in the park is 155 trees/ha which is not to dense because it is newly established park with small size of tree and canopy. Both deciduous and evergreen species were not significantly different in their average ΔTs (4.35 ± 0.40 °C vs. 4.38 ± 0.73 °C, *p*-value = 0.91) (Table 5). Moreover, the t-test did not indicate any significant differences in average leaf thickness, lightness of leaf color, and LAI between the deciduous and evergreen species (*p*-value > 0.05). Species that were able to greatly reduce the soil surface temperature or ΔTs (>5.00 °C) were *E. grandifloras*, *M. quinquenervia* and *A. lacucha*. The lowest ΔTs was estimated for *H. odorata* and *M. hortensis*. Both the highest and lowest ΔTs were estimated for the evergreen species (Table 5), suggesting that during the study period (cool dry season), when all the species are still foliated, they would reduce the soil temperature similarly and provide essential services to the park. The formula used to calculate ΔTs in this study is based on eq. 2, with a high weightage to the coefficient of leaf thickness (+3.775), indicating that thick leaves would reduce temperature more effectively by preventing heat penetration. *M. quinquenervia* and *A. lacucha* had the thickest leaf (>0.30 mm), indicating that a species with high LAI could moderate a higher change in soil temperature reduction compared to a species with lower LAI. However, even though *C. tabularis* had a high LAI (2.09 or the third highest in this study), its ability to reduce the soil temperature was limited due to very thin leaves (0.14 mm) (Table 5).

The annual average of reduction in air temperature (ΔTa) of the 21 species ranged from 0.94 °C to 3.28 °C (Table 6). The midday (10 a.m.–2 p.m.) ambient air temperature was between 34.05–37.71 °C. No significant difference in the average ΔTa was found between the deciduous (2.05 ± 0.58 °C) and evergreen species (1.86 ± 0.59 °C), respectively (t-test, *p*-value = 0.47) (Table 6). LAI contributed the most to the air-cooling effect (r = 0.65) with *B. purpurea*, *E. grandiflorus*, and *H. tomentosum* providing the greatest air-cooling due to their large, dense canopies and high LAI (Table 6).

A trade-off between air pollution tolerance and cooling effect indicates how well the plants can survive in such elevated pollution levels and be able to maintain vitality and provide optimum ecosystem services to human well-being in urban environments. We categorized the species into 4 groups based on a compromise between air pollution tolerance and the average degree of cooling based on the reduction in air and soil temperature, as indicated in Figure 1. The most recommended species, based on this criterion, included five evergreen (*M. quinquenervia*, *D. alatus*, *A. saman, S. indica*, and *C. tabularis*) and five deciduous (*A. lacucha*, *D. cochinchinensis*, *T. rosea*, *B. acutangular*, and *A. xylocarpa*). Two evergreen species (*H. tomentosum* and *M.hortensis*) are recommended to be planted on the roadside as they have a high pollution tolerance (as indicated by APTI in Table 4). The remaining nine species were more suitable to be planted in parks and recreation centers, given their ability to reduce the air and soil temperature, where shading and cooling are of primary concern. We did not find any species that could be classified under the “not recommended” category.

## 3. Discussion

Air pollution and UHI effects pose a continued risk to human health as well as to the health and vitality of species. Plants growing in urban areas are expected to remove air pollution as well as provide shade. Previous studies have focused only on specific plant traits that can simultaneously absorb pollutants [32,33]. Plants tolerant to pollution can thrive in such environments and help to absorb pollution from various anthropogenic sources. However, sensitive plants (such as *H. tomentosum* in this study) might not be able to withstand elevated pollution levels and would eventually deteriorate, but such species can be used as indicators of high air pollution levels [34]. Only a few studies have reported on how urban plants can survive and be healthy in conditions frequently found in urban areas [35].

APTI and API are indices which can be used as tools to indicate how well a species can tolerate air pollution and if they would be suitable for such environments [36]. Such studies can help urban planners and policy makers to decide on appropriate species which can provide ecosystem services contributing to human well-being in urban areas. In this study, the most recommended species (mostly evergreen), given the prevailing conditions in CU100 park, are suggested in Figure 1 based on a high API and cooling effect. Evergreen species maintain green leaves throughout the year and provide shading as well as absorb air pollution, especially PM_2.5_. Such species tend to grow continuously regardless of the season and have a better light adaptability [37], due to a smaller leaf size and dense canopy. The recommended species best suited to be planted as street trees were those with a high API but low cooling. On the other hand, given their distant location from the sources of pollution, parks or resting areas can have species that provide high ambient cooling but a relatively lower tolerance to pollution.

APTI is widely used to indicate the level of species tolerance to pollutants. In the present study, species with a high APTI largely had a high ascorbic acid content, such as *A. saman*, *M. quinquenervia*, *C. tabularis*, and *D. alatus*, resulting in a high API as well. We observed that maintaining an alkaline leaf pH level contributed the most to a high APTI. An alkaline pH may increase the conversion efficiency of hexose sugar to ascorbic acid and hence, it is directly related to a high APTI and API [38].

In this study, most species were categorized as being highly tolerant pollution according to the classification recommended by Shannigrahi, Fukushima and Sharmaa [9]. However, using the classification scheme of other studies [8,24] would have resulted in the species reported in this study as having intermediate tolerance or even sensitive [8]. In summary, we conclude that the sample species in CU100 Park have an intermediate to high pollution tolerance. In particular, the APTI values of *A. saman*, *M. hortensis*, *B. purpurea*, and *T. bellirica* were between 12–13 in this study, while the APTI of the same species growing in urban areas of India, was reported between 11–17 by Pandey [24]. The higher reported ATPI range for the species growing in India could be due to higher air pollution levels in India relative to Thailand as well as differences in plant age and stage of growth. This could also indicate that the same species has the ability to adapt to environments with higher air pollution levels. This observation was also made by Rai and Panda [39], who reported that plants growing in areas experiencing higher air pollution levels had a higher APTI than those growing in areas with lower levels. Moreover, seasonal variations can also affect pollution concentration and APTI. Das and Prasad [40] reported that APTI levels were higher in winter (December) compared to rainy and summer seasons due to the air inversion effect. We conducted the experiment during the months of November to January, which is classified as dry cool season and the highest measured levels of pollution during the year. However, as indicated by the data, the air pollution level was moderate at best (Table 3), which could have resulted in a lower APTI (Table 2) even during the cooler months.

Species that are pollution tolerant (as indicated by API) along with the morpho-biological traits and the economic value of the species can be considered during the planning of urban green spaces [41]. It has been previously reported that the dust-filtering ability of a plant species is correlated with the foliar surface characteristics [5,42] and includes characteristics such as the orientation relative to the main axis, size, area, shape, and surface [9,17]. We suggest that the traits that can be considered while selecting potential tree species for urban areas experiencing high levels of air pollution include year-round evergreen foliage (to absorb pollution), relatively smaller size and hairy leaves (to absorb pollutants), and a dense canopy with a high LAI (resulting in a high ATPI and API).

The ambient cooling effect was determined by the reduction in soil and air temperature (ΔTs and ΔTa). Soil temperature is as important as air temperature as it can affect the living organisms and microorganisms residing in the soil profile and plant roots [43]. Heat conduction, measured in terms of latent heat flux emanating from the ground, depends on the extent of paved surfaces, amount of soil moisture, and the penetration depth of solar radiation through a canopy [44]. The interaction between roots and microorganisms living in the root zone can enhance the growth and development of plants, and is also influenced by soil temperature [45]. The optimum temperatures for tropical plant species varies between 15–35 °C [46], with the root function hampered for temperatures outside this range. Therefore, maintaining soil temperature within the prescribed range is necessary, especially in urban areas where bare soil is exposed to direct sunlight. Moreover, anthropogenic activities such as transportation, built pavements, and burning can generate enough heat to affect the temperature of the soil profile. By providing shade, tree canopies can help to reduce the soil surface temperature. Lin and Lin [7] reported a reduction in soil temperature, ΔTs, between 3.28 °C to 8.07 °C in a Taiwanese Park with a similar tropical plant species profile, as reported in the present study (3.56 °C to 5.90 °C). The maximum ΔTs estimated in our study was lower than that reported by Lin and Lin [7] due to a lower LAI range (0.54–3.20 in the present study vs. 1.40–6.11). Peters and McFadden [47] indicated that dense tree canopies (LAI = 6) can reduce surface temperatures by up to 6 °C relative to sparse canopies with a near-zero LAI and further suggested that the soil temperature would reduce with increasing LAI.

Canopy cover, as measured by LAI, can play an important role in the reduction of ambient air and soil temperatures. A study in the Suzhou industrial park in Shanghai, China reported a highly significant positive relationship (*p*-value < 0.01, R^2^ = 0.915) between cooling effect and LAI [48]. Their observations were similar to this study in that the species with a relatively large stem size and dense canopy generally has a higher ΔTa and ΔTs, as seen for species like *B. purpurea* and *E. grandifloras* (Table 5), while the lowest reduction was estimated for small sized species with a relatively low LAI, such as *D. alatus, T. rosea* and *E. hygrophilus* (Table 3 and Table 4). Vidrih and Medved [49] calculated ΔTa in an old park (>50 years) with an average LAI of 3.16 and reported that the trees were able to reduce the air temperature by up to 4.8 °C, which is higher than that estimated in the present study (0.94 °C to 3.28 °C). The observed difference could be due to smaller trees with a younger age profile and a lower LAI (Table 6) in our study.

Plants, especially big trees, can regulate the local air and soil temperatures by reflecting a major fraction of the incident solar radiation and shading the soil surface, in addition to increasing the water vapor content via transpiration. Unlike plants, most reflective surfaces in urban areas such as streets, buildings, artificial fields etc. absorb and release heat [50], leading to an increase in ambient temperature, especially during the cool dry season (Dec–Jan), caused by air inversion. This trapped air could have resulted in the highest measured midday ambient temperature (37.71 °C) (Figure 2), leading to a large difference between the air temperature under the tree canopy and reference temperature during December and January, resulting in a high ΔTa. During the hot dry season (Mar–May), ΔTa was lowest probably due to higher levels of solar radiation and dry air leading to stomatal closure. This would result in reduced transpiration to reduce the difference between leaf and air temperature [51].

While considering planting of trees for cooling, the important traits to focus at are the lower leaf thickness and higher leaf relative water content to prevent heat penetration. Leaves of lighter color (see L* column in Table 5) reflect more and absorb less heat compared to darker leaves. Tree height is unlikely to be as important as the crown width when determining the cooling effect, as suggested by Speak et al. (2020) [52]. Low light penetration under dense canopies (high LAI), and hence less air and soil heat absorption under such canopies [5,53]. Trees of large size with deeper roots and higher leaf area can transpire large amounts of water, which can potentially lower the ambient air temperature compared smaller trees.

## 4. Materials and Methods

The study area was conducted in the Chulalongkorn University Park or CU100 Park (13.73′ N, 100.52′ E), with the local climate being classified as tropical monsoonal (Figure 3). The Park was designed specifically for climate change resilience with its major ecological service being prevention of inland flooding. Within a total area of 4.48 ha, balled and burlapped plants (56 species) were established in the Park with a mature canopy, mainly to avoid surface runoff from frequent flooding [6,54]. The wet season spans from May to October and the region experiences dry season from November to April with an average temperature between 32–34 °C. The average measured rainfall is between 1400–1600 mm with average relative humidity between 60–80% [55]. The air quality index (AQI) in the park area during the study period (dry season during November 2018 to January 2019) was measured between 22 and 200, with average levels hovering around 51 (classified as low to moderate) (Table 3).

### 4.1. Tree Selection

The plant inventory in the Park was reported by Yarnvudhi, Leksungnoen, Tor-Ngern, Premashthira, Thinkampheang and Hermhuk [6], with a total of 697 individual sampled trees, classified into 56 species in 49 genera and 22 families. We selected only those species which had at least 5 individuals (to be used as replicates), resulting in a total of 21 species, to determine the APTI, API, and the reduction in air (ΔTa) and soil (ΔTs) temperatures (Table 1). The required permissions to study the Park were granted by the Property Management of Chulalongkorn University. The voucher specimens were collected and identified by an expert from Faculty of Forestry (Dr. Dokrak Marod), and compared with the specimens listed in the Bangkok Forestry Herbarium (BKF), Thailand, according to the national guidelines listed by the Forest Herbarium Department of National Parks, Wildlife and Plant Conservation. All the species specimens identified in this study were then stored in the Department of Biology Herbarium, Faculty of Forestry, Bangkok. With regards to leaf phenology, 11 species were deciduous, while 10 species were evergreen. Most of the selected species are native to Thailand with only two species being exotic and native to South America (*Albizia saman* and *Tabebuia rosea*). The diameter at breast height (DBH), total height, crown area, and leaf area index (LAI) were measured for all the selected individuals. LAI, which is the ratio of area of leaves to the ground area under the canopy, was measured using a leaf area index meter (LAI-2200C, Li-Cor Inc., Lincoln, Nebraska, USA).

### 4.2. Calculation of Air Pollution Tolerant Index (APTI) and Anticipated Performance Index (API)

Several stresses such as large swings in temperature, high irradiation, drought, and salinity as well as air pollution can cause the production of reactive oxygen species (ROS) in plant cells, leading to oxidative stress/damage [56,57,58,59,60]. ROS are relatively short lived, unstable, and highly reactive molecules, possessing unpaired valence shell electrons resulting in the disruption of cellular homeostasis, leading to cell damage and eventual cell death [61,62]. Ascorbic acid is a natural antioxidant produced by plants to increase tolerance to ROS [40]. Therefore, species with higher levels of ascorbic acid would be expected to have a relatively higher tolerance to air pollution.

APTI and API were calculated during the cool dry season from November 2018 to January 2019, as the highest pollution levels are measured during this time of the year, with no rain to leach out the pollutants from the air and leaves. For each of the selected 21 species, five individual trees of similar size and health were selected (Table 1). In each individual tree, two mature, fully expanded sunlit leaves per individual tree (210 leaf samples in total) were collected. Furthermore, the second or third leaf from an accessible shoot was selected to avoid too old or too young leaves from being sampled. All leaf samples were collected between 6 am and 8 am LT to ensure they were devoid of any mid-day stress. Leaf samples were then swiftly wrapped in plastic bags and then put in an ice box covered with aluminum foil to minimize the water lost through transpiration. The box was transferred to the laboratory within an hour to prevent any degradation of leaf samples. In the laboratory, samples were washed with distilled water to clean any dust particles and dried with a tissue paper before any measurement was undertaken.

To calculate the APTI, ascorbic acid content, total chlorophyll content, leaf extract pH, and relative leaf water content were measured in fresh samples using the formula [8],
(1)$$\mathrm{APTI}=\frac{\left[A\times \left(T+P\right)\right]+R}{10}$$
where *A* is ascorbic acid content (mg g^−1^); *T* is total chlorophyll content (mg g^−1^); *P* is leaf extract pH; *R* is relative water content of leaf (percent), respectively.

The estimated APTI value was thus used to indicate the tolerance level of a species to air pollution using the classification recommended by Shannigrahi, Fukushima and Sharmaa [9] as follows; APTI < 7 is sensitive, APTI 10–11 is intermediate, and APTI > 12 is tolerant. APTI was then used to calculate the anticipated performance index (API) according to Shannigrahi, Fukushima and Sharmaa [9]. It was estimated using APTI and relevant biological and economic variables including stem size, canopy structure, leaf phenological type, leaf structure, and economic value. API was determined based on a character grading scale (+ or −) ranging from 7 (recommended) to 0 (not recommended) (Table 2). For example, we estimated the APTI of *Melaleuca quinquenervia* as 12.261, giving it +++++ grade, as it is a large tree (++), with a spreading dense canopy (++), evergreen (+), medium (+), coriaceous (+), hardy lamina structure (+), and has a high economic value (++). Hence, the species got a total score was 15 out of a maximum 16 (94%), giving it an API score of 7 (Table 4).

### 4.3. Calculation of Reduction in Soil Temperature (ΔTs)

The reduction in soil temperature (ΔTs) under the canopy of the selected species was calculated with the same five individuals per species, during the cool dry season, from November 2018 to January 2019. In each individual tree, five leaves were used to calculate the ΔTs using a stepwise multiple regression equation purposed by Lin and Lin [7],
ΔTs (°C) = 9.186 + 0.655 LAI + 3.755 leaf thickness (mm) + 0.643 leaf texture + 0.025 L* − 3.267 vapor pressure (kPa) + 0.682 solar radiation (mW. M^−2^) − 0.063 wind velocity (Km.h^−1^)(2)
where LAI is Leaf area index (LAI) (unitless) and was measured in four directions for each selected tree during 6–9 am and 4–6 pm to avoid strong and direct sunlight of the day (mostly at 10 am–2 pm in Thailand). The leaf thickness was measured using a 0.001 mm-resolution leaf thickness gauge (547–401, Mitutoyo Corporation, Kawasaki, Japan), with the leaf texture of the upper leaf surface being classified as either smooth (0) or rough (1), as indicated by visual and sensory observation. L* is the lightness coefficient of a leaf and ranges from dark = 0 to light = 100 and was measured with a color meter (MiniScan EZ, HunterLab Inc., Reston, Virginia, USA). The hourly values of vapor pressure, solar radiation, and wind velocity were obtained from a weather station located 16 km away from the study site.

### 4.4. Measurement of Reduction in Air Temperature (ΔTa)

ΔTa was measured during November 2018–December 2019 as the difference in temperatures between the under-tree canopy and outside the canopy in full sunlight (reference temperature). Automatic temperature sensors (Elitech RC-5, Elitech UK.Ltd, London, UK) were used to measure the air temperature and were programmed to collect the data every minute for the entire study period. Twenty-one sensors were placed under the canopy of the selected species and a temperature data logger was placed on the tree stem at a height of 1.5 m from the ground facing north to prevent any interference from direct solar radiation.

We measured one individual tree per species for each of the 21 species in order to monitor the temperature (one minute resolution) over the whole year. Trees were chosen based on similar DBH, height, crown cover, and LAI to minimize any bias related to tree size (see t-test in Table 6). In order to calculate the ΔTa, air temperature measured by each sensor was averaged daily between 10 am to 2 pm (when temperature was the highest). Temperature difference between the reference temperature and that under the canopy of each tree was then averaged for the entire year (see Table 6). A statistical analysis of ΔTa among the 21 species was conducted through a one-way ANOVA using daily ΔTa as replicates using the R statistical program (Version R4.0.3, R Development Core team, 2020).

## 5. Conclusions

The structure and function of trees in parks are key to provide optimum ecosystem services. Appropriate species selection is critical and must be planned according to a park’s environment. The decision should focus on services related to cooling effect from trees as being the priority, but it should be supplemented with whether or not the selected species can survive air pollution (as indicated by ATPI and API). Both evergreen and deciduous species can provide ambient cooling but can be vulnerable to air pollution, such as *E. grandiflorus* and *B. purpurea.* On the other hand, some species can have a higher air pollution tolerance but fail to provide adequate cooling, such as *H. odorata* and *M. hortensis*. Therefore, based on our study, *A. lacucha*, *D. cochinchinensi*, *M. quinquenervia*, and *A. saman* are the most recommended species, which are tolerant to air pollution and can provide shading to reduce the air and soil temperature. Results would be beneficial for urban greenspace planners in tropical monsoonal regions while selecting the most suitable species to be planted in urban green spaces that can provide services like air pollution mitigation and temperature reduction to humans without reducing plant vitality, growth, and development.

## Figures and Tables

**Figure 1 plants-11-03074-f001:**
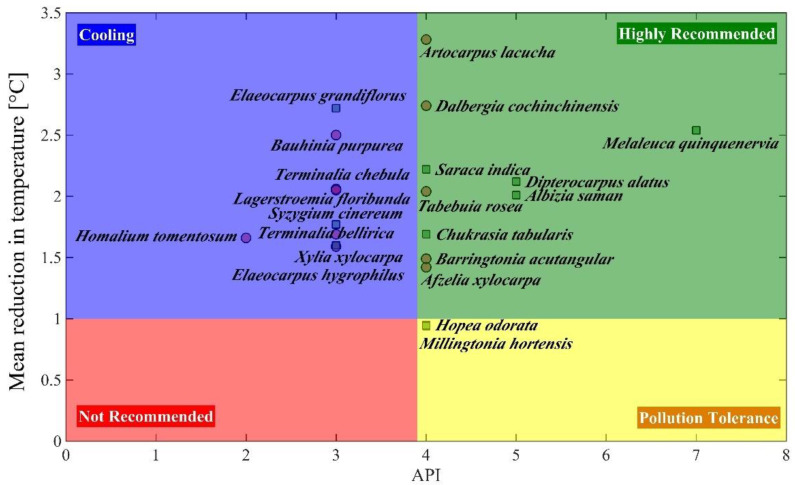
Species recommendations for compromising between air pollution tolerance and cooling effect. The filled-squares indicate evergreen species while filled-circles indicate deciduous species.

**Figure 2 plants-11-03074-f002:**
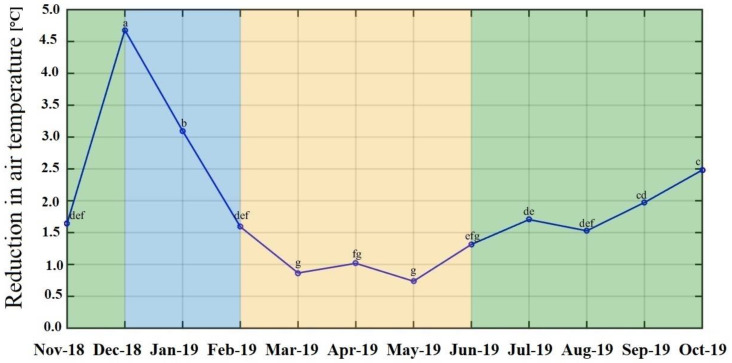
Average monthly reduction in air temperature of the selected 21 species in the CU100 Park. The averages were calculated for data measured over a span of 1 year from November 2018 to October 2019. The blue, yellow, and green patches indicate the months classified under the cool dry, hot dry, and rainy seasons, respectively. The lowercase letters on the line indicate the statistical difference in the mean reduction of air temperature at a level of 95% (*p*-value < 0.05).

**Figure 3 plants-11-03074-f003:**
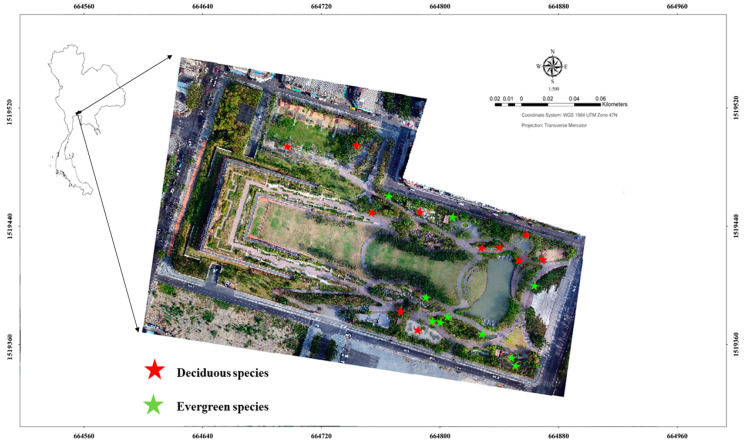
Location of the Chulalongkorn University Centenary Park (CU 100 Park) in Thailand, Bangkok metropolitan area with 21 evergreen and deciduous trees’ species examined in an area of 4.48 ha.

**Table 1 plants-11-03074-t001:** The selected 21 species in the CU100 Park.

No.	Species	Forest Type	Habitat	Count (Number of Trees)	LAI	Crown Cover (m^2^)	DBH (cm)	Total Height (m)	Voucher Specimen Name
1	*Dalbergia cochinchinensis* Pierre	D	Native	136	2.51 ± 5.29	4.57 ± 2.21	13.82 ± 2.11	8.33 ± 1.43	CU01-2018
2	*Tabebuia rosea* (Bertol.) Bertero ex A. DC.	D	Exotic	81	1.05 ± 0.42	2.78 ± 1.40	12.71 ± 2.42	6.73 ± 0.87	CU02-2018
3	*Albizia saman (Jacq.) Merr.*	E	Exotic	74	1.32 ± 0.44	15.75 ± 7.42	19.70 ± 4.89	7.71 ± 1.42	CU03-2018
4	*Millingtonia hortensis* L. f.	E	Native	70	1.36 ± 0.58	1.99 ± 0.88	14.50 ± 1.78	7.97 ± 1.21	CU04-2018
5	*Dipterocarpus alatus* Roxb. ex G. Don	E	Native	47	0.93 ± 0.49	1.62 ± 0.94	12.57 ± 2.25	8.19 ± 1.38	CU05-2018
6	*Afzelia xylocarpa* (Kurz) Craib	D	Native	27	5.76 ± 25.17	2.37 ± 1.19	13.54 ± 2.03	7.26 ± 0.83	CU06-2018
7	*Hopea odorata* Roxb.	E	Native	26	2.15 ± 0.88	1.85 ± 1.66	13.68 ± 2.14	8.70 ± 1.37	CU07-2018
8	*Homalium tomentosum* (Vent.) Benth.	D	Native	23	1.32 ± 0.44	3.03 ± 1.47	12.41 ± 1.74	8.86 ± 1.56	CU08-2018
9	*Lagerstroemia floribunda* Jack var. floribunda	D	Native	22	1.72 ± 0.52	2.74 ± 1.03	12.9 ± 2.35	6.45 ± 0.95	CU09-2018
10	*Bauhinia purpurea* L.	D	Native	20	2.47 ± 1.03	5.18 ± 2.05	16.28 ± 23.41	6.01 ± 1.10	CU10-2018
11	*Xylia xylocarpa* (Roxb.) W. Theob. var. kerrii (Craib & Hutch.) I. C. Nielsen	D	Native	17	1.71 ± 0.54	3.22 ± 1.39	13.74 ± 2.33	8.47 ± 1.22	CU11-2018
12	*Syzygium cinereum (L.) Skeels*	E	Native	12	1.55 ± 0.94	2.72 ± 1.29	16.98 ± 1.99	6.45 ± 0.26	CU12-2018
13	*Melaleuca quinquenervia* (Cav.) S.T. Blake	E	Native	11	1.36 ± 0.39	2.23 ± 0.92	17.20 ± 2.88	8.73 ± 1.60	CU13-2018
14	*Chukrasia tabularis* A. Juss.	E	Native	9	1.13 ± 0.61	2.42 ± 1.06	13.49 ± 3.12	7.69 ± 0.90	CU14-2018
15	*Terminalia chebula* Retz. var. chebula	D	Native	8	1.21 ± 0.46	1.98 ± 0.92	13.44 ± 3.47	6.89 ± 1.24	CU15-2018
16	*Baringtonia acutangula* (L.) Gaertn.	D	Native	7	0.95 ± 0.69	2.70 ± 1.74	17.01 ± 3.11	5.96 ± 0.86	CU16-2018
17	*Saraca indica* L.	E	Native	7	0.87 0.21	2.50 0.53	11.77 ± 1.21	4.56 ± 0.61	CU17-2018
18	*Elaeocarpus hygrophilus* Kurz	E	Native	7	1.04 ± 0.51	1.84 ± 0.73	10.50 ± 2.98	5.20 ± 1.05	CU18-2018
19	*Terminalia bellirica* (Gaertn.) Roxb.	D	Native	6	0.69 ± 0.19	3.10 ± 1.21	14.48 ± 1.60	9.70 ± 0.70	CU19-2018
20	*Artocarpus lacucha* Roxb. ex Buch.-Ham.	D	Native	5	2.07 ± 0.52	2.61 ± 0.65	14.08 ± 0.87	8.79 ± 1.25	CU20-2018
21	*Elaeocarpus grandiflorus* Sm.	E	Native	5	1.78 ± 0.92	1.99 ± 0.87	7.86 ± 2.38	3.28 ± 0.62	CU21-2018

Remarks: Forest type: “D” = Deciduous, “E” = Evergreen. Habitat: “A” = Central/North/South America, “NA” = native to Thailand. Forest type: “D” = Deciduous, “E” = Evergreen. Mean ± standard deviation as determined from the tree count of each species.

**Table 2 plants-11-03074-t002:** Gradation of plant species based on the air pollution tolerance index (APTI) and other biological and socio-economic characteristics of anticipated performance index (API) of the plant species (modified from Shannigrahi et al., 2004).

Grading Characteristic			Pattern of Assessment	Grade Allotted
(a) Tolerance	Air pollution tolerance index (APTI)		7.0–8.0	+
			8.1–10.0	++
			10.1–11.0	+++
			11.1–12.0	++++
			12.1–13.0	+++++
(b) Biological and socio-economic	(i) Stem size		Small	-
			Medium	+
			Large	++
	(ii) Canopy structure		Spare/Irregular/Globular	-
			Spreading crown/open/semi-dense	+
			Spreading dense	++
	(iii) Leaf phenological type		Deciduous	-
			Evergreen	+
	(iv) Leaf structure	Size (length × width)	Small (2–5 × 1.5–3 cm)	-
			Medium (5–15 × 3–8 cm)	+
			Large (8–25 × 5–15 cm)	++
		Texture	Smooth (no hair)	-
			Coriaceous (hairy)	+
		Hardiness	Delineate (easy to tear)	-
			Hardy (leathery)	+
	(iv) Economic value		Less than three uses	-
			Three or four uses	+
			Five or more uses	++
After the calculation of APTI, an assessment of the anticipated performance index (API) of plant species (Maximum grade that can be scored by an individual = 16).				
**Grade**	**Score (%)**		**Assessment of plant species**	
0	Up to 30		Not recommended for planting	
1	31–40		Very poor	
2	41–50		Poor	
3	51–60		Moderate	
4	61–70		Good	
5	71–80		Very good	
6	81–90		Excellent	
7	91–100		Best	

**Table 3 plants-11-03074-t003:** Concentration of air pollutants in CU100 Park with the standard air quality.

Pollutants	Study Area	WHO (2021)	Thailand’s Standard	Pollution Level
PM_2.5_ (μg/m^3^)	38.01 ± 25.27	15	25	High
PM_10_ (μg/m^3^)	49.24 ± 32.36	45	50	Moderate
NO_2_ (ppb)	22.47 ± 12.99	13	60	Moderate
SO_2_ (ppb)	2.72 ± 1.36	15	100	Low
CO (ppm)	0.90 ± 0.34	4	4.4	Low
O_3_ (ppb)	19.38 ± 14.03	51	35	Low

**Table 4 plants-11-03074-t004:** APTI and species evaluation based on APTI values, in addition to relevant biological and economic characteristics (Shannigrahi et al., 2004).

No	Plant Species	Total APTI	Assessment Parameters										
			APTI	Tree Habit	Canopy Structure	Type of Tree	Laminar Structure			Economic Importance	Grade Allotted		API Grade
							Size	Texture	Hardiness		Total Plus (+)	% Scoring	
1	*Melaleuca quinquenervia* (E)	12.261 ± 0.06 ^bc^	+++++	++	++	+	+	+	+	++	15	94	7
2	*Dipterocarpus alatus* (E)	12.130 ± 0.06 ^bcdce^	+++++	++	-	+	++	+	+	++	14	88	6
3	*Albizia saman* (E)	13.059 ± 0.16 ^a^	+++++	++	++	+	-	-	-	++	12	75	5
4	*Chukrasia tabularis* (E)	12.191 ± 0.10 ^bcd^	+++++	++	++	+	+	-	-	+	12	75	5
5	*Artocarpus lacucha* (D)	11.455 ± 0.06 ^fghi^	++++	++	++	-	+	+	-	++	12	75	5
6	*Saraca indica* (E)	11.441 ± 0.12 ^fghi^	++++	++	++	+	+	-	+	+	12	75	5
7	*Baringtonia acutangular* (D)	11.805 ± 0.06 ^cdef^	++++	+	++	-	+	-	+	++	11	69	4
8	*Afzelia xylocarpa* (D)	11.667 ± 0.09 ^efg^	++++	++	++	-	+	-	-	++	11	69	4
9	*Tabebuia rosea* (D)	11.662 ± 0.16 ^efgh^	++++	++	++	-	+	+	+	-	11	69	4
10	*Lagerstroemia floribunda* (D)	11.820 ± 0.13 ^cdef^	++++	+	++	-	++	+	+	-	11	69	4
11	*Hopea odorata* (E)	11.479 ± 0.09 ^fghi^	++++	++	++	-	+	-	-	++	11	69	4
12	*Millingtonia hortensis* (E)	12.380 ± 0.11 ^b^	+++++	++	++	+	-	-	-	-	10	63	4
13	*Dalbergia cochinchinensis* (D)	10.348 ± 0.23 ^k^	+++	++	++	-	-	+	-	++	10	63	4
14	*Bauhinia purpurea* (D)	11.809 ± 0.07 ^cdef^	++++	+	++	-	+	-	-	++	10	63	4
15	*Syzygium cinereum* (E)	11.707 ± 0.06 ^def^	++++	+	++	+	+	-	-	+	10	63	4
16	*Terminalia bellirica* (D)	11.420 ± 0.10 ^fghij^	++++	++	++	-	+	-	-	+	10	63	4
17	*Elaeocarpus hygrophilus* (E)	11.205 ± 0.11 ^ghij^	++++	+	++	+	+	-	-	+	10	63	4
18	*Terminalia chebula* (D)	11.179 ± 0.07 ^hij^	++++	++	++	-	+	-	-	+	10	63	4
19	*Elaeocarpus grandiflorus* (E)	11.161 ± 0.10 ^ij^	++++	-	++	+	+	+	-	+	10	63	4
20	*Xylia xylocarpa* (D)	10.946 ± 0.11 ^j^	+++	++	++	-	+	-	-	++	10	63	4
21	*Homalium tomentosum* (D)	11.677 ± 0.08 ^efg^	++++	++	++	-	+	-	-	-	9	56	3

Remarks: Forest type “D” = deciduous species, “E” = evergreen species. Different letters in the total APTI column indicate a statistically significant difference in mean at a significant level of 95%

**Table 5 plants-11-03074-t005:** Soil temperature reduction under canopy of 21 species from November 2018 to January 2019.

Species	Soi Temperature Reduction (°C)	Leaf Thickness	L*	LAI	Roughness
**Deciduous**					
*Artocarpus lacucha*	5.02 ± 0.49 ^c^	0.31 ± 0.04	33.71 ± 1.32	1.36 ± 0.36	hairy
*Dalbergia cochinchinensis*	4.76 ± 0.61 ^d^	0.15 ± 0.01	34.54 ± 3.56	1.83 ± 0.71	hairy
*Bauhinia purpurea*	4.64 ± 0.50 ^de^	0.19 ± 0.02	38.47 ± 2.20	2.24 ± 0.35	smooth
*Lagerstroemia floribunda*	4.62 ± 0.47 ^de^	0.23 ± 0.05	39.46 ± 1.66	0.96 ± 0.40	hairy
*Terminalia chebula*	4.50 ± 0.55 ^ef^	0.27 ± 0.04	44.33 ± 2.76	1.36 ± 0.45	smooth
*Tabebuia rosea*	4.38 ± 0.46 ^fg^	0.25 ± 0.02	36.81 ± 4.22	0.60 ± 0.12	hairy
*Terminalia bellirica*	4.30 ± 0.77 ^g^	0.26 ± 0.03	39.74 ± 6.05	1.30 ± 0.81	smooth
*Homalium tomentosum*	3.94 ± 0.51 ^h^	0.25 ± 0.07	43.24 ± 4.82	0.65 ± 0.32	smooth
*Xylia xylocarpa*	3.94 ± 0.56 ^h^	0.17 ± 0.03	38.02 ± 1.92	1.30 ± 0.55	smooth
*Barringtonia acutangular*	3.88 ± 0.54 ^hi^	0.22 ± 0.02	39.92 ± 4.99	0.88 ± 0.47	smooth
*Afzelia xylocarpa*	3.85 ± 0.59 ^hi^	0.16 ± 0.02	40.61 ± 2.09	1.16 ± 0.59	smooth
**Average**	**4.35 ± 0.40**	**0.22 ± 0.05**	**38.99 ± 3.23**	**1.24 ± 0.49**	
**Evergreen**					
*Elaeocarpus grandiflorus*	5.90 ± 0.55 ^a^	0.27 ± 0.02	41.28 ± 5.84	2.65 ± 0.58	hairy
*Melaleuca quinquenervia*	5.32 ± 0.57 ^b^	0.33 ± 0.05	32.29 ± 6.95	1.74 ± 0.35	hairy
*Saraca indica*	4.36 ± 0.45 ^fg^	0.23 ± 0.03	47.26 ± 3.13	1.26 ± 0.13	smooth
*Dipterocarpus alatus*	4.34 ± 0.46 ^fg^	0.20 ± 0.02	45.06 ± 2.23	0.54 ± 0.15	hairy
*Albizia saman*	4.30 ± 0.58 ^g^	0.26 ± 0.02	33.04 ± 2.06	1.55 ± 0.62	smooth
*Syzygium cinereum*	4.29 ± 0.53 ^g^	0.28 ± 0.02	38.41 ± 3.04	1.20 ± 0.37	smooth
*Chukrasia tabularis*	4.28 ± 0.57 ^g^	0.14 ± 0.02	36.66 ± 5.27	2.09 ± 0.51	smooth
*Elaeocarpus hygrophilus*	3.74 ± 0.48 ^ij^	0.25 ± 0.02	39.59 ± 5.02	0.50 ± 0.12	smooth
*Hopea odorata*	3.68 ± 0.45 ^jk^	0.19 ± 0.02	44.16 ± 2.51	0.61 ± 0.08	smooth
*Millingtonia hortensis*	3.56 ± 0.53 ^k^	0.14 ± 0.03	34.45 ± 2.98	1.03 ± 0.63	smooth
**Average**	**4.38 ± 0.69**	**0.23 ± 0.06**	**39.22 ± 4.94**	**1.31 ± 0.67**	
** *t-test (p-value)* **	** *0.91 (NS)* **	** *0.83 (NS)* **	** *0.37 (NS)* **	** *0.77 (NS)* **	

Remarks: “L*” = lightness of leaf color (with 0 indicating dark and 100 indicating light). Different letters in the total APTI column indicate a statistically significant difference in the mean values at a significant level of 95%. The t-test is compared between the means of deciduous and evergreen species. NS indicates no significant difference in the mean.

**Table 6 plants-11-03074-t006:** Reduction in air temperature under the canopy of 21 tree species during the period from November 2018 to October 2019.

Species	Air Temperature Reduction (°C)	LAI	Crown Cover (m)	DBH (cm)	Total Height (m)
**Deciduous**					
*Bauhinia purpurea*	3.28 ± 1.77 ^a^	2.76	4.4	9.6	6.7
*Homalium tomentosum*	2.74 ± 1.94 ^ab^	1.19	4.1	10.5	9.2
*Lagerstroemia floribunda*	2.50 ± 1.71 ^abc^	1.17	2.9	13.5	4.1
*Artocarpus lacucha*	2.06 ± 1.41 ^bcd^	1.33	3.4	15.6	8.4
*Baringtonia acutangula*	2.05 ± 1.04 ^bcd^	1.67	3	16.4	5.5
*Dalbergia cochinchinensis*	2.04 ± 1.39 ^bcd^	0.6	4.9	14	7.9
*Terminalia bellirica*	1.69 ± 0.99 ^bcd^	1.71	2.1	13.1	10.1
*Afzelia xylocarpa*	1.66 ± 1.34 ^bcd^	0.61	4	13.6	9.6
*Tabebuia rosea*	1.59 ± 1.44 ^cd^	0.61	3.5	13.9	8.8
*Xylia xylocarpa*	1.49 ± 1.46 ^cd^	1.22	2.7	13.6	8.9
*Terminalia chebula*	1.42 ± 1.37 ^cd^	1.05	1.2	10.2	6
**Average**	**2.05 ± 0.58**	**1.27 ± 0.63**	**3.29 ± 1.07**	**13.09 ± 2.16**	**7.75 ± 1.91**
**Species**	**Air temperature reduction (°C)**	**LAI**	**Crown cover (m)**	**DBH (cm)**	**Total height (m)**
**Evergreen**					
*Elaeocarpus grandiflorus*	2.72 ± 1.07 ^ab^	3.2	2.5	11.9	3.3
*Melaleuca quinquenervia*	2.54 ± 1.16 ^abc^	1.48	1.8	15.3	11
*Albizia saman*	2.22 ± 1.30 ^abc^	0.69	21	18	6.4
*Chukrasia tabularis*	2.12 ± 1.29 ^abcd^	1.51	2.8	13.1	7.9
*Syzygium cinereum*	2.01 ± 1.33 ^bcd^	0.77	1.1	17.4	8.5
*Hopea odorata*	1.77 ± 1.35 ^bcd^	0.54	1.3	12.3	8.8
*Millingtonia hortensis*	1.69 ± 1.10 ^bcd^	1.29	1.7	14.7	9.2
*Saraca indica*	1.60 ± 1.39 ^bcd^	1.04	3.2	12.8	2.8
*Dipterocarpus alatus*	0.95 ± 1.42 ^d^	0.68	3.2	14.9	9.2
*Elaeocarpus hygrophilus*	0.94 ± 1.33 ^d^	0.59	2.8	12.1	6.4
**Average**	**1.86 ± 0.60**	**1.18 ± 0.80**	**4.14 ± 5.97**	**14.25 ± 2.19**	**7.35 ± 2.64**
** *t-test (p-value)* **	** *0.47 (NS)* **	** *0.78 (NS)* **	** *0.65 (NS)* **	** *0.24 (NS)* **	** *0.70 (NS)* **

Remarks: The different letters in the total APTI column indicates a statistically significant difference in mean at a significance level of 95%. The t-test compares the means of the deciduous and evergreen species. NS indicates no significant difference in the mean.

## Data Availability

Not applicable.

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
