# Peer review of "Assessing the Cooling and Air Pollution Tolerance among Urban Tree Species in a Tropical Climate"

_plants, 2022, doi:10.3390/plants11223074_

Round 1
Reviewer 1 Report
The review concerned an article entitled: Balancing Cooling and Air Pollution Tolerance among Urban Tree Species in a Tropical Climate. The topic mentioned in this article is very important this days, especially in urbanized locations. Also the air pollution becomes more dangerous for human and vegetation. In my opinion this article touches very important issue.
Because I’m not an English native speaker I want be checking the manuscripts language.
Nevertheless I have some questions and comments:
Abstract – In my opinion some results should be add.
Line 52 – I have doubts about this sentence. Cities are very different. In the different places in earth results and differences in temperature can be higher.
Line 57 – Authors do not mention about the traffic and cars which are in most cities the first cause of PM air pollution.
Line after 75 – I think some sentences about the accumulation of the PM should be add here. There is a lack of it, but we have many articles on this topic.
Line 80 – There is no citation to prove this sentence. For eg. DOI:10.15244/pjoes/78626 is talking about this topic.
Line 94. The full Latin name of the species always should have the name of author. Also in other parts of manuscript.
Line 101-104 – I think this is part of conclusions. It should be at the end of manuscript.
Line 119 – PM should be written with small 10 and 2.5 – PM10 and PM2.5
Line 138. Double dot.
Line 140. I think the numbering of tables is wrong.
Line 152. Author should describe the exact position of trees inside the park. The trees in the center of park have different conditions with pollution. Also other trees and shrubs can shade each other. There were some ponds, lakes, fountains in the park – the breeze can also affect the results in my opinion.
Line 222. The PM2.5
Line 262-263. Not only the leaves characteristics – also very important is the composition of plants. It was already described in some papers as. DOI:10.3390/su14052973 or DOI:10.1016/j.scitotenv.2021.147310.
Line 324 – The name of cited authors should be add.
Line 345 – the PM should be written with small 10 and 2.5 – PM10 and PM2.5
Line -367-378 – This part is more related to the introduction not the methodology.
Author Response
General Reply
The authors would like to thank the editor and the reviewers for their constructive comments that have made the manuscript more concise and readable. The changes have been marked in the revised manuscript named plants-1938274 (the color red has been used to highlight the responses to the reviewer questions).
Reviewer comments
The review concerned an article entitled: Balancing Cooling and Air Pollution Tolerance among Urban Tree Species in a Tropical Climate. The topic mentioned in this article is very important this days, especially in urbanized locations. Also the air pollution becomes more dangerous for human and vegetation. In my opinion this article touches very important issue.
Because I’m not an English native speaker I want be checking the manuscripts language.
Answer: The manuscript has been proofread.
Abstract
P 2 Line 52: I have doubts about this sentence. Cities are very different. In the different places in earth results and differences in temperature can be higher.
Answer: The reviewer is right in pointing out the different characteristics of each city. However, in the paragraph, we intended to provide the number of how urban heat island effect in the city that had been reported. According to the suggestion, we have added the name of the city and the country we are talking about. Currently on page 2 line 52-56.
P2 Line 57: Authors do not mention about the traffic and cars which are in most cities the first cause of PM air pollution.
Answer: The changes have been made as per the suggestion as “Anthropogenic activity not only causes UHIs but can also result in air pollution, especially from construction, industrial production, agricultural stubble burning, traffic, and cars, increasing the levels of particulate matter (PM) (Popek et al., 2022)”. Currently on page 2 line 58-59.
P2 Line after 75: I think some sentences about the accumulation of the PM should be add here. There is a lack of it, but we have many articles on this topic.
Answer: The changes have been made as per the suggestion as “Species with hairs and a thick waxy coating are the effective characteristics to accumulate PM. Trees in the center of the park would retain PM in longer time when compare to those at the edge or close to the road because of dense tall canopy acting as the insulting layer for air pollution (Przybysz et al., 2021)” Currently on page 2 line 80-83.
P2 Line 80: There is no citation to prove this sentence. For eg. DOI:10.15244/pjoes/78626 is talking about this topic.
Answer: The citation of Popek et al., 2018 has been added as per the suggestion. Currently on page 2 line 86.
P2 Line 94: The full Latin name of the species always should have the name of author. Also in other parts of manuscript.
Answer: The Latin name with the name of the author of the species have been added throughout the manuscript.
P 2-3 Line 101-104: I think this is part of conclusions. It should be at the end of manuscript.
Answer: The sentence “Results would be beneficial for urban greenspace planners in tropical monsoonal regions while selecting the most suitable species to be planted in urban green spaces that can provide services like air pollution mitigation and temperature reduction to humans without reducing plant vitality, growth, and development.” has been moved to page 18-19 line 474-478.
P3 Line 119: PM should be written with small 10 and 2.5 – PM10 and PM2.5
Answer: The “PM10 or PM2.5” have been revised to PM10 and PM2.5. Currently on page 3 line 132.
P3 Line 138: Double dot.
Answer: The “double dot” has been revised to “single dot.” Currently on page 3 line 150.
P4 Line 140: I think the numbering of tables is wrong.
Answer: The numbering of tables have been adjusted throughout the manuscript. Currently on page 4, 6, 8, 9, 15, and 17 on lines 152, 158, 186, 199, 361, and 427.
P7 Line 152: Author should describe the exact position of trees inside the park. The trees in the center of park have different conditions with pollution. Also other trees and shrubs can shade each other. There were some ponds, lakes, fountains in the park – the breeze can also affect the results in my opinion.
Answer: We indicate the location of the tress as per suggestion through the sentences “The exact position of the trees in the park was shown in the Fig. 1. Most of the trees were distant from the road in approximately the same (as in Fig. 1, main road are around the park). The density of the trees in the park is 155 trees/ha which is not too dense because it is newly established park with small size of tree and canopy.” Currently on page 7 line 165-169.
P11 Line 222: The PM2.5
Answer: The “PM2.5” has been revised to “PM2.5” Currently on page 11 line 237.
P12 Line 262-263: Not only the leaves characteristics – also very important is the composition of plants. It was already described in some papers as. DOI:10.3390/su14052973 or
DOI:10.1016/j.scitotenv.2021.147310.
Answer: The changes have been made as per the suggestion to “It has been previously reported that the dust-filtering ability of plant species is correlated with the foliar surface characteristics (Popek et al., 2022; Govindaraju et al., 2011) and includes characteristics such as orientation to the main axis, size, area, shape, surface, and economic value (Shannigrahi et al., 2004; Przybysz et al., 2021)”. Currently on page 12 line 277-279.
P14 Line 324: The name of cited authors should be add.
Answer: The changes have been made as per the suggestion to “Low light penetration occurs under dense canopies (high LAI), and hence less air and soil heat absorption under such canopies (Popek et al., 2022; Huber et al., 2021).”. Currently on page 14 line 339-341.
P15 Line 345: the PM should be written with small 10 and 2.5 – PM10 and PM2.5
Answer: The “PM10 or PM2.5” have been revised to PM10 and PM2.5 Currently on page 15 line 362.
P15 Line -367-378: This part is more related to the introduction not the methodology.
Answer: The part previously in lines 367-378 “Under stresses induced by air pollution, a highly acidic (low pH) environment can build up in the cell sap due to acidic gases (SO2, NO2, or CO2) in the ambient air forming acid radicals in the leaf tissue (Pandey, 2015; Sumangala et al., 2018). Such acidic gases can damage the cell membrane after entering through the stomata (Khanoranga and Phytomonitoring, 2019). Therefore, species that can maintain alkaline levels would be more tolerant to air pollution. Relative water content (R) is associated with cell turgor pressure and protoplasmic permeability (Karmakar et al, 2021) with high levels diluting acidity inside the leaf cell sap, resulting in resistance to air pollution (Karmakar et al, 2021; Kumar and Nandini, 2013; Kaur and Nagpal, 2017). Kumar and Nandini, 2013 indicated that stress caused by air pollution can decrease the chlorophyll content in species. As chlorophyll content is directly related to the photosynthesis process and in turn to the growth and development of plants, high chlorophyll content would be directly related to high growth and air pollution tolerance.” has been moved to pages 2-3 in lines 93-103.

Reviewer 2 Report
General comments:
The study and the written article are valuable contributions to the selection of different species of “right” tree to the urban areas. The trees that can thrive in stressful urban areas are also potential candidates to mitigate heat island effect and air pollution. The results and discussion are convincing and well-written. However, it is just the abstract which should be amended to tally with the rest of the article.
Specific comments:
Topic:
“Balancing”
Comment: the word “balancing” seems to anticipate a relationship between the “cooling” and “air pollution tolerance”, but I do not see such correlation between the two. It seems to be that the article is about “Assessing urban tolerance and cooling effect”
Abstract:
Line 29: “mitigate”
Comment: “tolerance” and “mitigation” are two levels of language taxonomy. Since I do not see the study addresses the “mitigation” part of the assessed trees, it should only be “tolerance” (assessed by APTI and API). Maybe I am wrong, but the utility of API is not clearly explained. Can API indirectly infer the “mitigation” ability?
Line 33: “Malelueca” should be “Melaleuca”
Line 34: “and was able” should be “and were able”
Line 38: “Buhinia” should be “Bauhinia”
Keywords:
Delete “air pollution tolerance” if it already mentioned in the title
Introduction:
Line 96: “can mitigate”--- again, you only assessed the “tolerance”, not “mitigate”.
Please also go through by using search tools to see if you have mistakenly mixed up the word “tolerance” and “mitigate” unless you have seriously addressed “how trees remove or filter the pollutants”. In Line 260, you seemed to draw the correlation between API and “efficient absorption of air pollution (by the way, it should be pollutants)”, but you have not explained clearly how API can infer the absorption ability in the “introduction” part. Please consider adding the definition of API and how it can infer the “mitigation” ability in the introduction (rather than merely “classify” the level of “pollution tolerance” (Line 65)). If you can draw such correlation, the whole idea of “mitigation” would be alright.
Author Response
General Reply
The authors would like to thank the editor and the reviewers for their constructive comments that have made the manuscript more concise and readable. The changes have been marked in the revised manuscript named plants-1938274 (the color red has been used to highlight the responses to the reviewer questions).
Reviewer comments
General comments:
The study and the written article are valuable contributions to the selection of different species of “right” tree to the urban areas. The trees that can thrive in stressful urban areas are also potential candidates to mitigate heat island effect and air pollution. The results and discussion are convincing and well-written. However, it is just the abstract which should be amended to tally with the rest of the article.
Topic:
P1 Line2: Comment: the word “balancing” seems to anticipate a relationship between the “cooling” and “air pollution tolerance”, but I do not see such correlation between the two. It seems to be that the article is about “Assessing urban tolerance and cooling effect”
Answer: “cooling” has been revised to “assessing”. Currently on page 1 line 2.
Abstract:
P1 Line 29: “tolerance” and “mitigation” are two levels of language taxonomy. Since I do not see the study addresses the “mitigation” part of the assessed trees, it should only be “tolerance” (assessed by APTI and API). Maybe I am wrong, but the utility of API is not clearly explained. Can API indirectly infer the “mitigation” ability?
Answer: “mitigation” has been revised to “tolerance”. Currently on page 1 line 29.
P1 Line 33: “Malelueca” should be “Melaleuca”
Answer: “Malelueca” has been revised to “Melaleuca”. Currently on page 1 line 33.
P1 Line 34: “and was able” should be “and were able”
Answer: “and was able” has been revised to “and were able”. Currently on page 1 line 34-35.
Line 38: “Buhinia” should be “Bauhinia”
Answer: “Buhinia” has been revised to “Bauhinia”. Currently on page 1 line 39.
Keywords:
P1 Line 43-44: Delete “air pollution tolerance” if it already mentioned in the title
Answer: “air pollution tolerance” has been deleted as per the suggestion.
Introduction:
P2 Line 96: “can mitigate” again, you only assessed the “tolerance”, not “mitigate”.
Answer: “mitigate” has been revised to “tolerance”. Currently on page 3 line 113.
Please also go through by using search tools to see if you have mistakenly mixed up the word “tolerance” and “mitigate” unless you have seriously addressed “how trees remove or filter the pollutants”.
Answer: Changes have been made from “mitigate” to “tolerance or tolerant” at relevant places in the manuscript.
P12 Line 260: you seemed to draw the correlation between API and “efficient absorption of air pollution (by the way, it should be pollutants)”, but you have not explained clearly how API can infer the absorption ability in the “introduction” part. Please consider adding the definition of API and how it can infer the “mitigation” ability in the introduction (rather than merely “classify” the level of “pollution tolerance” (P2 Line 65). If you can draw such correlation, the whole idea of “mitigation” would be alright.
Answer: The changes have been made as per the suggestion to “Species that are pollution tolerant (as indicated by API) along with the morpho-biological traits and the economic value of the species can be considered during the planning of urban green spaces”. Currently on page 12 line 274-276.

Reviewer 3 Report
1. Looking at the papers obtaining the APTI value, it can be seen that the range is different, and the APTI value may vary depending on which paper is referenced, so a detailed review is needed.
ex) Prajapati. K. and Tripathi (2008) Anticipated performance index of some tree species considered for green belt development in and around an urban area: A case study of Varanasi city, India, Journal of Environmental Management 88:1343–1349.
Pattern of Assessment 9.0-12.1 +, 12.1-15.0 ++, 15.1-18.0 +++, 18.1-21.0 ++++, 21.1-24.0 +++++
Saadullah Khan Leghari et al. (2019) Estimating anticipated performance index and air pollution tolerance index of some trees and ornamental plant species for the construction of green belt, Pol. J. Environ. Stud. 28(3):1759-1769.
Pattern of Assessment 01-04 --, 05-08 -, 09-12 +, 13-16 ++, 17-20 +++, 21-24 ++++, 25-28 +++++
Author Response
General Reply
The authors would like to thank the editor and the reviewers for their constructive comments that have made the manuscript more concise and readable. The changes have been marked in the revised manuscript named plants-1938274 (the color red has been used to highlight the responses to the reviewer questions).
Reviewer
Comments and Suggestions for Authors
- Looking at the papers obtaining the APTI value, it can be seen that the range is different, and the APTI value may vary depending on which paper is referenced, so a detailed review is needed.
- ex) Prajapati. K. and Tripathi (2008) Anticipated performance index of some tree species considered for green belt development in and around an urban area: A case study of Varanasi city, India, Journal of Environmental Management 88:1343–1349.
Pattern of Assessment 9.0-12.1 +, 12.1-15.0 ++, 15.1-18.0 +++, 18.1-21.0 ++++, 21.1-24.0 +++++
Saadullah Khan Leghari et al. (2019) Estimating anticipated performance index and air pollution tolerance index of some trees and ornamental plant species for the construction of green belt, Pol. J. Environ. Stud. 28(3):1759-1769.
Pattern of Assessment 01-04 --, 05-08 -, 09-12 +, 13-16 ++, 17-20 +++, 21-24 ++++, 25-28 +++++
Answer: We thank the reviewer for the posed question. We also concluded that many ranges of the APTI have been reported and appreciate the suggestion from the reviewer. Below is a summary table of the ranges of APTI.
Grade allotted (APTI) |
Shannigrahi et al., 2004 from West Bangal India (Used in this manuscript) |
Grade allotted (APTI) |
Prajapati et al., 2008, Varnasi,India, (where pollution exceed limit) |
Grade allotted (APTI) |
Leghari et al., 2019 (Pakistan) |
+ |
7.0-8.0 |
+ |
9.0-12.0 |
-- |
01-04 |
++ |
8.1-10.0 |
++ |
12.1-15.0 |
- |
05-08 |
+++ |
10.1-11.0 |
+++ |
15.1-18.0 |
+ |
09-12 |
++++ |
11.1-12.0 |
++++ |
18.1-21.0 |
++ |
13-16 |
+++++ |
12.1-13.0 |
+++++ |
21.1-24. |
+++ |
17-20 |
|
|
|
|
++++ |
21-24 |
|
|
|
|
+++++ |
25-28 |
The reason we used Shannigrahi et al., 2004 in this manuscript is because the weather and pollution levels were in a similar range as our study area. Moreover, the range and level of APTI as well as the species composition were similar to the present study. For example, Saraca indica in our study has an APTI of 11.44 while in Shannigrahi et al., 2004 reported a value of 9.70.
In Prajapati et al., 2008 and Leghari et al., 2019, the maximum APTI was 24 and 28, respectively, while the maximum APTI in our study was 13 which was also reported as the maximum value by Shannigrahi et al., 2004. Additionally, the APTI range in our study was 10 – 13. The binning range reported by Shannigrahi et al., 2004 was narrower when compared to the other two, leading to a higher resolution of the change in APTI and the subsequent grade allotted, especially when the values are in a narrower range as is the case for our study (10-13). Therefore, if we had used the range reported by either Prajapati et al., 2008 or Leghari et al., 2019, every species would have been assigned a grade of + or ++ with none of the species having a +++ grade.
Round 2
Reviewer 3 Report
The authors have adequately revised the manuscript.